# Biopsy-Controlled Non-Invasive Quantification of Collagen Type VI in Kidney Transplant Recipients: A Post-Hoc Analysis of the MECANO Trial

**DOI:** 10.3390/jcm9103216

**Published:** 2020-10-07

**Authors:** Manuela Yepes-Calderón, Camilo G. Sotomayor, Daniel Guldager Kring Rasmussen, Ryanne S. Hijmans, Charlotte A. te Velde-Keyzer, Marco van Londen, Marja van Dijk, Arjan Diepstra, Stefan P. Berger, Morten Asser Karsdal, Frederike J. Bemelman, Johan W. de Fijter, Jesper Kers, Sandrine Florquin, Federica Genovese, Stephan J. L. Bakker, Jan-Stephan Sanders, Jacob Van Den Born

**Affiliations:** 1Division of Nephrology, Department of Internal Medicine, University Medical Center Groningen, University of Groningen, 9713 AV Groningen, The Netherlands; manueyepes@gmail.com (M.Y.-C.); r.s.hijmans@umcg.nl (R.S.H.); c.a.keyzer@umcg.nl (C.A.t.V.-K.); m.van.londen@umcg.nl (M.v.L.); m.van.dijk02@umcg.nl (M.v.D.); s.p.berger@umcg.nl (S.P.B.); s.j.l.bakker@umcg.nl (S.J.L.B.); j.sanders@umcg.nl (J.-S.S.); j.van.den.born@umcg.nl (J.V.D.B.); 2Nordic Bioscience A/S, 2730 Herlev, Denmark; dgr@nordicbio.com (D.G.K.R.); MK@nordicbioscience.com (M.A.K.); fge@nordicbio.com (F.G.); 3Department of Pathology and Medical Biology, University Medical Center Groningen, University of Groningen, 9713 AV Groningen, The Netherlands; a.diepstra@umcg.nl; 4Department of Nephrology, Amsterdam University Medical Center, University of Amsterdam, 1105 AZ Amsterdam, The Netherlands; f.j.bemelman@amsterdamumc.nl; 5Department of Nephrology, Leiden University Medical Center, University of Leiden, 2300 RC Leiden, The Netherlands; J.W.de_Fijter@lumc.nl; 6Amsterdam Institute for Infection and Immunity (AII), Amsterdam UMC, University of Amsterdam, 1098 XH Amsterdam, The Netherlands; j.kers@amsterdamumc.nl (J.K.); s.florquin@amsterdamumc.nl (S.F.); 7Amsterdam Cardiovascular Sciences (ACS), Amsterdam UMC, University of Amsterdam, 1098 XH Amsterdam, The Netherlands; 8Leiden Transplant Center, Department of Pathology, Leiden University Medical Center, 2300 RC Leiden, The Netherlands; 9Van ‘t Hoff Institute for Molecular Sciences (HIMS), University of Amsterdam, 1098 XH Amsterdam, The Netherlands

**Keywords:** kidney transplantation, fibrosis, inflammation, extracellular matrix, collagen type VI

## Abstract

The PRO-C6 assay, a reflection of collagen type VI synthesis, has been proposed as a non-invasive early biomarker of kidney fibrosis. We aimed to investigate cross-sectional and longitudinal associations between plasma and urine PRO-C6 and proven histological changes after kidney transplantation. The current study is a post-hoc analysis of 94 participants of the MECANO trial, a 24-month prospective, multicenter, open-label, randomized, controlled trial aimed at comparing everolimus-based vs. cyclosporine-based immunosuppression. PRO-C6 was measured in plasma and urine samples collected 6 and 24 months post-transplantation. Fibrosis was evaluated in biopsies collected at the same time points by Banff interstitial fibrosis/tubular atrophy (IF/TA) scoring and collagen staining (Picro Sirius Red; PSR); inflammation was evaluated by the tubulo-interstitial inflammation score (ti-score). Linear regression analyses were performed. Six-month plasma PRO-C6 was cross-sectionally associated with IF/TA score (Std. β = 0.34), and prospectively with 24-month IF/TA score and ti-score (Std. β = 0.24 and 0.23, respectively) (*p* < 0.05 for all). No significant associations were found between urine PRO-C6 and any of the biopsy findings. Fibrotic changes and urine PRO-C6 behaved differentially over time according to immunosuppressive therapy. These results are a first step towards non-invasive fibrosis detection after kidney transplantation by means of collagen VI synthesis measurement, and further research is required.

## 1. Introduction

Kidney transplantation is the best available treatment for patients with end-stage kidney disease [1,2]. In recent times, short-term graft survival has seen great improvement, which unfortunately has not been paralleled by equivalent improvement in long-term graft survival [3]. An important threat to long-term graft survival is progressive loss of kidney allograft function related to progressive fibrosis [4]. Despite its clinical importance, early identification of fibrosis appearance remains a challenge [5]. Currently, biopsy samples are the gold standard for the detection of established kidney fibrosis, but this has the evident drawback as a follow-up measurement of requiring an invasive procedure, which generates discomfort for the patients and can be complicated by bleeding. Other drawbacks are sampling variability and sampling errors [5,6]. Therefore, great interest exists in finding non-invasive biomarkers that can detect fibrosis formation, ideally at early stages [4].

Kidney allograft fibrosis reflects a pathological response to injury where the equilibrium between extracellular matrix formation and degradation is deregulated and progressive deposition of collagens, among other matrix constituents, takes place [7,8]. Assessment of active collagen formation may identify kidney transplant recipients (KTRs) at high risk of fibrosis progression and therefore development of chronic graft failure [9,10]. Among the different collagens, collagen type VI (COL VI) is found in the kidney and is constantly produced by fibroblasts at relative low levels in the interstitium, the intima and adventitia layers of the kidney vasculature, as well as in the glomeruli [11,12,13]. Under normal conditions, COL VI has an important physiological role in maintaining extracellular matrix (ECM) structure and function, controlling matrix and cell orientation [14]. However, under pathological conditions (e.g., chronic kidney disease), its active deposition in the kidneys is massively increased [9,12]. During production of COL VI, the C5 domain at the C-terminal of the α3 chain is released from the immediate pericellular matrix [15]. The PRO-C6 assay detects the C-terminal end of this domain and is proposed as a surrogate biomarker for COL VI active formation [9]. Moreover, the cleavage of part of this domain gives rise to a bioactive molecule, named endotrophin, which is also detected by the PRO-C6 assay [15,16]. Endotrophin has important biological effects, such as attracting macrophages, increasing transforming growth factor-β (TGFβ) signaling, promoting epithelial–mesenchymal transition, adipose tissue fibrosis, and metabolic dysfunction [17]. Increased plasma levels of PRO-C6 have previously been associated with the progression of chronic kidney disease and, specifically in the post-transplantation setting, with reduced graft function in KTR [4,9,18,19]. Whether associations between PRO-C6 and decreased graft function indeed correspond to increased fibrotic or inflammatory changes in the kidneys and whether it could be used as a non-invasive biomarker for fibrosis development in KTR remain unknown.

In the current study, we aimed to investigate the cross-sectional and longitudinal associations between PRO-C6 in plasma and urine, and proven histological changes in KTR of the minimization of maintenance immunosuppression early after kidney transplantation (MECANO) trial, which is a randomized, controlled, open-label, multicenter trial testing early cyclosporine A (CsA) elimination. Furthermore, since it is known that CsA nephrotoxicity includes pathological increased production and decreased degradation of extracellular matrix proteins, including collagen, and TGF-β up-regulation [20,21,22], we explored a potential differential role of PRO-C6 as a biomarker of fibrosis among patients under different immunosuppressive regimens.

## 2. Materials and Methods

### 2.1. Study Design and Population

Between November 2005 and June 2009, 361 de novo KTRs were recruited in three Dutch transplantation centers to participate in the MECANO trial (trial registration: NTR1615). The study was conducted according to the Good Clinical Practice guidelines, in accordance with the ethical principles of the Declaration of Helsinki, and was approved by the Dutch Medical Ethical Board for medical research (METC 04/154, 1 October 2004) [23,24]. All patients signed written informed consent forms. This study was a 24-month, prospective, multi-center, open-label, randomized, controlled trial, aiming at optimizing maintenance immunosuppression and reducing side effects. During the first six months after enrollment, all patients had a similar quadruple immunosuppressive regimen: induction with basiliximab, followed by CsA, mycophenolate sodium (MPS), and prednisolone [24]. At month six, a protocol biopsy was performed. When no histological signs of rejection were seen, patients were randomized to receive dual immunosuppressive therapy with CsA (*n* = 89), MPS, or everolimus (EVL) (*n* = 96), all in combination with prednisolone. In case of (borderline) rejection patients, were not randomized. The primary endpoint of the MECANO study was the development of interstitial fibrosis at the 24-months protocol biopsy.

After enrollment of 39 patients, the MPS-group was prematurely stopped by the Data Safety Monitoring Board because of an unacceptably high rejection percentage (21%). The trial continued as a two-group trial, comparing CsA and EVL. The results of the primary outcome of the study were published in 2016 [23].

### 2.2. Protocol Kidney Biopsies and Histological Analyses

Protocol biopsies were scheduled at 6 and 24 months after transplantation. At six months, biopsies were obtained in 99% and 98% of patients in the CsA group and the EVL group, respectively. Of the available biopsies, 78% and 81% in the CsA group and the EVL group were considered adequate, respectively. At 24 months, biopsies were obtained in 84% and 79% of patients in the CsA group and the EVL group, respectively. The prevalence of adequate samples was 81% and 73% in the CsA group and the EVL group, respectively (*p* = 0.4, two-tailed). The current study reports the results of the 94 patients (51 in the CsA group and 43 in the EVL group) whose 6-month biopsies met the minimal adequacy threshold of seven glomeruli and one artery.

Tissues were formalin-fixed and paraffin-embedded and stained with periodic-acid Schiff diastase, hematoxylin/eosin, and Jones’ methenamine silver. Two independent kidney pathologists (Amsterdam University Medical Center (UMC) and Leiden UMC, The Netherlands), unaware of any clinical data, classified the biopsies according to the 2015 update of the Banff classification [25] and assigned a Banff interstitial fibrosis/tubular atrophy (IF/TA) score. Morphometric analysis of cortical interstitial fibrosis was centralized at the Amsterdam UMC. Adequate protocol biopsy sections were stained with Picro Sirius Red (PSR, Aldrich, Munich, Germany), which is used for the detection of collagen fibers. PSR-stained slides were digitalized using a slide virtual microscope system (Olympus, Tokyo, Japan) with a 20× magnification objective and saved in Tagged Image File Format (TIFF format). Image analyses were performed with the ImageJ software package (National Institutes of Health, Bethesda, MD, USA) where the PSR-stained area was aut omatically assessed by means of a macro. All input was verified manually. Inflammation was evaluated by the total percentage of inflamed cortical area (ti-score) as a continuous score [26].

### 2.3. PRO-C6 Detection

Plasma and urine PRO-C6 concentrations were measured using a competitive enzyme-linked immunosorbent assay (Nordic Bioscience, Herlev, Denmark) that specifically detects the last 10 amino acids of the alpha-3 chain of COL VI (3168′KPGVISVMGT3177′) and is validated for both sample matrices [27]. The assay has a detection limit of 0.15 ng/mL and a 95% confidence interval for inter- and intra-assay variability in plasma samples reported as 3.4%–12.4% and 1.1%–5.3%, respectively [19]. For urine samples, the detection limit was the same as plasma, and inter- and intra-assay variability are reported as 7.9% and 3.2%, respectively [9]. To account for variations in urine concentration, urinary PRO-C6 was divided by urinary creatinine, measured by the QuantiChrom™ Creatinine Assay Kit (BioAssay Systems, Hayward, CA, USA), and the PRO-C6/creatinine ratio was used in all analyses.

### 2.4. Statistical Analyses

Data analyses, computations, and graphs were performed with SPSS 25.0 software (IBM Corporation, Chicago, IL, USA). To test whether variables were normally distributed, a histogram was generated for each variable. For descriptive statistics data were presented as mean (standard deviation (SD)) for normally distributed data, and as median (interquartile range (IQR)) for variables with a non-normal distribution. Categorical data were expressed as number (percentage).

Differences in plasma and urine PRO-C6 and biopsy changes (IF/TA score, PSR, and ti-score) among subgroups of KTRs according to their treatment regimen and to their primary kidney disease were tested by one-way ANOVA for continuous variables with normal distribution, Mann–Whitney U test for continuous variables with skewed distribution, and X^2^ test for categorical variables. Linear regression analyses were performed to study the association of plasma and urine PRO-C6 with biopsy changes at 6 and 24 months and the delta between the two visits. Furthermore, subgroup analyses were performed by dividing patients by the immunosuppressive regimen used. We also performed sensitivity analyses, in which patients who were grouped under “unknown cause” as primary kidney disease were recoded as if they have been suffering from glomerulonephritis as primary kidney disease. For all statistical analyses, a 2-sided *p* < 0.05 was considered significant.

## 3. Results

### 3.1. Baseline Characteristics

The characteristics at enrollment and at randomization of a total of 94 patients, 51 in the CsA group and 43 in the EVL group, are displayed in Table 1 and Table 2. At enrollment, in the overall population, the mean (*SD*) age was 52 (13) years-old, and most patients were male and Caucasian. The main cause of end-stage kidney disease in this trial was polycystic kidney disease (24%), followed by glomerulonephritis (17%) and hypertension (16%). The mean donor age was 50 (13) years old, and the most frequent type of donors was living unrelated (31%), followed closely by deceased after brain death (30%). The median antigen mismatch was 3, and the median (IQR) of total time on kidney replacement therapy was 24 (5–46) months.

At randomization, 6 months after the beginning of the trial, patients had a mean graft function, as assessed by the estimated glomerular filtration rate (eGFR), of 49 (42–62) mL/min/1.73 m^2^. Patients had a mean weight of 79 (14) kg and a mean systolic blood pressure of 144 (20) mmHg. Mean low-density lipoprotein (LDL) was 3.19 (2.39–3.75) mmol/L, and 59% of patients were statin users. Mean glycated hemoglobin was 6.08% (1.10), and only two patients had the diagnosis of diabetes mellitus. Fifteen patients (16%) were active smokers. Concerning subgroup differences, patients in the CsA group had an apparent higher weight (81 vs. 78 kg), a more frequent use of statins (63% vs. 54%), and a higher percentage were active smokers (20% vs. 12%) when compared to the EVL group. Also, the two diabetic patients were both in the EVL group. None of these differences was of statistical significance.

### 3.2. PRO-C6 and Biopsy-Proven Histological Changes over Follow-Up

Mean (*SD*) plasma PRO-C6 at 6 and 24 months was 9.5 (3.4) and 9.4 (4.3) ng/mL, respectively, without significant differences between the two groups. As for urine, median (IQR) PRO-C6 at 6 and 24 months after correction by creatinine was 6.7 (4.8–12.4) and 5.9 (3.4–21.5) ng/mg, respectively. Plasma and urine PRO-C6 did not correlate at either 6 or 24 months (Spearman’s ρ 0.226, *p* = 0.09; Spearman’s ρ 0.311, *p* = 0.11; respectively). No difference in urine PRO-C6 between the two study groups was present at 6 months, but at 24 months mean urine PRO-C6 was significantly higher in the EVL group compared to the CsA group (7.5 vs. 4.5 ng/mg; *p* = 0.02). Delta plasma PRO-C6 was positive in both subgroups and was not significantly different. As for delta urine PRO-C6, it was positive in the EVL group and negative in the CsA group; this difference was statistically significant (0.9 vs. −1.4 ng/mg; *p* = 0.01). (Table 3).

Histological analyses at 6 months showed a median IF/TA score of 1 (0–1) points and a mean PSR staining percentage of 13.3% (6.0), with no significant differences between patients in the CsA and EVL groups. Inflammation, as evaluated by the ti-score, was also not significantly different between the two groups. At 24 months, the overall population showed a higher IF/TA score, PSR percentage, and ti-score when compared to the previous biopsy. At this time point, the PSR staining percentage was higher in the CsA group compared to the EVL group (19.7% vs. 14.5%; *p* = 0.02); no significant difference was present in the other histological parameters (Table 3).

When patients were stratified by their primary kidney disease, no significant differences were found in the plasma and urine concentrations of PRO-C6 at any time point during follow-up. No significant difference was found either in fibrosis (IF/TA and PRO-C6) or inflammation (ti-score) at 6 and 24 months (Appendix A). Also, no significant differences were found in sensitivity analyses in which all KTRs with unknown cause of primary kidney disease were considered as patients with glomerulonephritis as primary kidney disease (Appendix A).

### 3.3. Association between PRO-C6 and Biopsy Changes

Plasma PRO-C6 at 6 months post transplantation was significantly associated with 6-month and 24-month IF/TA scores (Std. β = 0.34 and 0.24, respectively; both *p* < 0.05). A prospective association was also present for 6-month plasma PRO-C6 with 24-month biopsy proven inflammation (ti-score) and the delta inflammation between the two biopsies (Std. β = 0.23 and 0.22, respectively; both *p* < 0.05). No association was found between 6-month plasma PRO-C6 and 6- or 24-month PSR. Also, no cross-sectional association was found between 24-month plasma PRO-C6 and histological evidence of fibrosis or inflammation. Urine PRO-C6 at 6 months only showed a prospective and inverse association with 24-month PSR (Std. β = −0.30; *p* < 0.05), and there were no cross-sectional associations at 24 months. Delta plasma and urine PRO-C6 did not correlate with either histological evidence of fibrosis or inflammation (Table 4).

When patients were divided by randomization group, no significant associations were found between 24-month plasma PRO-C6 and histological changes. Urine PRO-C6 was significantly and inversely associated with the delta of IF/TA score in patients among the CsA group, and no other significant association was found. Delta plasma and urine PRO-C6 were not significantly associated with any histological changes (Table 5).

## 4. Discussion

This study shows, in a homogeneous well-characterized cohort of KTRs who were participants of the MECANO clinical trial, that 6-month post-transplant plasma concentration of PRO-C6 associates with graft biopsy-proven fibrotic and inflammatory changes, both cross-sectionally (IF/TA score) and longitudinally (IF/TA score and ti-score). Further, we show that these same associations are not found with 6-month urine PRO-C6, and that at 24 months, no cross-sectional association was present between fibrotic changes and either urine or plasma PRO-C6. Subgroup analyses comparing patients under CsA vs. EVL immunosuppressive therapy showed higher urinary concentration of PRO-C6 in the EVL group compared to the CsA groups during follow-up, despite lower fibrosis scorings.

The progression of kidney diseases is characterized by the appearance of progressive fibrosis, which reflects a pathological disequilibrium between the synthesis and degradation of ECM constituents, including collagens, within scarred kidneys [8,28]. COL VI is an ECM molecule distributed in the kidney interstitium, vasculature, and in the glomeruli, which is constantly produced by fibroblasts at relative low levels [12,13]. Under healthy conditions, it has an important physiological role in maintaining structure and function of the ECM by controlling organization and cell orientation [14]. However, its markedly increased synthesis and deposition has been reported under a wide spectrum of kidney pathologies [29,30].

COL VI biosynthesis and assembly involves a complex multi-step pathway [14,31]. During active deposition in the ECM, a pro-peptide in the α3 chain of COL VI is released; in turn, this gives rise to the bioactive molecule endotrophin [15,27], which is known to have a role in shaping a pro-inflammatory and pro-fibrotic microenvironment by, amongst other processes, triggering an increase in cytokines such as TGFβ [16]. The PRO-C6 assay measures both the release of endotrophin and of the pro-peptide, reflecting newly formed molecules of mature COL VI [9,27]. In the post-transplantation setting, the assessment of active collagen formation has been proposed as a way of early identifying KTRs that are at high risk of fibrosis progression [9,10], and since allograft function loss is closely related to the appearance and progression of interstitial fibrosis and tubular atrophy [10,32,33], it could identify also KTRs at future risk of developing chronic graft failure [9,10].

Clinically, increased deposition of COL VI has been reported in multiple scenarios of chronic kidney disease [28,31], and specifically in the post-transplantation setting, a strong association was found between increased plasma concentration of PRO-C6 and a decrease in graft function over time [4]. In agreement with this evidence, we found a positive prospective association between 6-month PRO-C6 concentration and biopsy evidence of increased graft fibrosis (IF/TA). However, no associations were found with PSR staining. Following the evidence that patients receiving CsA are at risk of developing nephrotoxicity, which is also a condition with unregulated ECM deposition and TGF-β upregulation [20,21,22,34], we performed exploratory analysis by subgroups of immunosuppressive therapy. When dividing the population into subgroups, we found that patients in the CsA group had higher PSR% at 24 months, but urine PRO-C6 was higher in the EVL group.

This analysis shows that PRO-C6 measurement, as reflection of collagen VI synthesis, is associated with, but not identical to, quantification of fibrosis in transplanted kidneys, especially not under different treatment conditions. The next considerations should be taken into account: first, by measuring plasma or urine PRO-C6, the cells/tissues where the existing collagen VI synthesis takes place cannot be identified and might be (partially) different from the transplanted kidney. Second, PRO-C6, by definition, only measures a collagen split product of the alpha3 chain of collagen VI [15,27], whereas PSR staining is the resultant of all collagen deposits. As we know, there are >20 different types of collagens, all of which can be stained by PSR [35]. So, changes in PSR staining do not necessarily correspond with changes in COL VI synthesis. Third, the PRO-C6 assay measures a split product of collagen VI that is cleaved off after cellular synthesis and thus reflects synthesis of collagen VI. Collagen deposition in a tissue, however, is the resultant of collagen synthesis and collagen degradation (mainly by metalloproteinases). So, the PRO-C6 assay shows one side of the coin (synthesis), whereas the other side of the coin (degradation) is not measured. We anticipate that various treatment regimens might not only influence collagen VI synthesis but collagen VI degradation as well. Next, since we did not perform immunofluorescent studies, we cannot assure that there was recurrence or enhanced interstitial inflammation; however, when stratified analyses by primary kidney disease were performed, there was no significant difference in biomarkers or histological evidence of inflammation. Also, the possibility that incidence of glomerulonephritis was underestimated due to low use of immunofluorescence in the evaluation of biopsy materials in the regular clinical setting in which the current study was performed, and the possibility that such potential underestimation may have biased our results, is a limitation of our study. Although we performed sensitivity analyses in which we found no indication of the presence of such bias, it can, of course, not be excluded. Future studies are warranted to confirm our findings, and it would be relevant to apply immunofluorescence in such studies in order to maximize the accuracy of estimation of glomerulonephritis recurrence. It would also be interesting if future studies would compare the pre- and post-transplantation behavior of PRO-C6. Furthermore, patients receiving CsA had a more marked decline in eGFR compared to the EVL group, as was shown in the main outcomes of the MECANO publication [23]. This might have influenced both plasma and urine PRO-C6 values and differences between both treatment arms. Finally, we do not have information on eGFR at inclusion, therefore the eGFR changes before randomization could not be evaluated, and this prevents us from exploring the causes underlying early fibrotic lesions.

The present study has several strengths. Being a randomized clinical trial, we have a very homogenous population regarding time since transplantation and initial immunosuppressive regimen. Also, we studied PRO-C6 against the current gold standard for fibrosis detection, which is kidney biopsy [5,6], taken at the same time point as the biomarkers, allowing both cross-sectional and longitudinal analyses. Several limitations must also be considered. Most of our patients are from a European background, and care should be taken when extrapolating our findings to other ethnic groups. Also, especially at 24 months, we had a reduced number of available samples and a longer follow-up would have allowed us to further explore the prospective behavior of PRO-C6.

In conclusion, 6-month post-transplantation plasma concentration PRO-C6 has a good longitudinal association with graft biopsy-proven IFTA scores, which could make it potentially useful as a follow-up tool. On the other hand, urine PRO-C6 did not associate with fibrotic parameters measured at time of biopsy or in future protocol biopsies. Additionally, we showed a differential evolution of PRO-C6 during follow-up dependent on immunosuppressive regimen. For the first time, this study provides biopsy-controlled data of PRO-C6 as a potential non-invasive biomarker of graft fibrosis in KTRs. This is a first step towards non-invasive detection by plasma PRO-C6 of pro-fibrotic ECM turnover early after transplantation. The potential utility of the implementation of PRO-C6 in clinical follow-up of KTRs requires further clinical studies.t The detection of causes underlying early kidney fibrosis was not the scope of the current study, yet we hold a plea for future studies aiming at evaluating whether primary kidney disease may influence he performance of PRO-C6 as a biomarker in KTRs. Furthermore, it would be interesting if future studies would also compare the pre- and post-transplantation behavior of PRO-C6.

## Figures and Tables

**Table 1 jcm-09-03216-t001:** Characteristics at enrollment of study population, overall kidney transplant recipients (KTRs), and randomization groups.

Characteristics at Enrollment	Overall	Randomized Group		*p* Value
		CsA	EVL	
Number of patients, *n*	94	51	43	
Age, years (SD)	52 (13)	51 (13)	54 (12)	0.30
Sex (male), *n (%)*	64 (68)	33 (65)	31 (72)	0.44
Race (Caucasian), *n (%)*	83 (88)	47 (92)	36 (84)	0.21
Primary kidney disease, *n (%)*				0.81
Polycystic kidney disease	24 (26)	13 (26)	11 (26)	
Glomerulonephritis	16 (17)	9 (18)	7 (16)	
Hypertension	15 (16)	7 (14)	8 (19)	
Urologic	8 (9)	3 (6)	5 (12)	
Vascular	5 (5)	2 (4)	3 (7)	
Focal segmental glomerulosclerosis	3 (3)	1 (2)	2 (5)	
Diabetes mellitus	3 (3)	2 (4)	1 (2)	
Unknown cause	16 (17)	11 (22)	5 (12)	
Donor type, *n (%)*				0.81
Living unrelated	29 (31)	15 (29)	14 (33)	
Deceased after brain death	28 (30)	15 (29)	13 (30)	
Living related	22 (23)	14 (28)	8 (19)	
Deceased after cardiac death	14 (15)	7 (14)	7 (16)	
Donor age, years (SD) ^a^	50 (13)	51 (13)	49 (12)	0.55
Antigen mismatch, *n* (IQR)	3 (2–4)	3 (2–3)	3 (2–4)	0.52
TTKRT, months (IQR)	24 (5–46)	18 (6–46)	24 (5–48)	0.53

^a^ Data available in 87 patients. CsA: cyclosporine A; EVL: everolimus; TTKRT: total time on kidney replacement therapy.

**Table 2 jcm-09-03216-t002:** Characteristics at randomization of study population, overall KTRs, and randomization groups.

Characteristics at Randomization	Overall	Randomized Group		*p* value
		CsA	EVL	
eGFR, mL/min/1.73 m^2^	49 (42–62)	49 (43–57)	49 (40–67)	0.89
Weight, kg (SD) ^a^	79 (14)	81 (15)	78 (13)	0.24
BMI, kg/m^2^ (SD) ^a^	26.7 (3.5)	26.0 (3.9)	25.4 (3.1)	0.44
SBP, mmHg (SD) ^b^	144 (20)	144 (20)	145 (21)	0.82
DBP, mmHg (SD) ^b^	84 (12)	84 (11)	83 (12)	0.68
LDL, mmol/L (SD) ^b^	3.19 (2.39–3.75)	3.19 (2.37–3.95)	3.15 (2.40–3.70)	0.84
HDL, mmol/L (SD) ^b^	1.39 (1.20–1.73)	1.30 (1.17–1.71)	1.49 (1.20–1.76)	0.49
Cholesterol, mmol/L (SD) ^b^	5.13 (4.34–6.10)	5.16 (4.26–6.23)	5.08 (4.40–6.07)	0.92
Statins use, *n (%)*	55 (59)	32 (63)	23 (54)	0.36
Glucose, mmol/L ^c^	5.10 (4.50–5.80)	5.10 (4.70–5.80)	4.90 (4.50–5.90)	0.35
HbA1c, % (SD) ^c^	6.08 (1.10)	6.14 (1.25)	6.01 (0.88)	0.60
Diabetes mellitus, *n (%)*	2 (2)	0 (0)	2 (5)	0.12
Smoking current, *n (%)*	15 (16)	10 (20)	5 (12)	0.29

Data available in ^a^ 90, ^b^ 92, and ^c^ 88 patients. eGFR: estimated glomerular filtration rate; BMI: body mass index; SPB: systolic blood pressure; DBP: diastolic blood pressure; LDL: low-density lipoprotein; HDL: high-density lipoprotein; HbA1c: glycated hemoglobin.

**Table 3 jcm-09-03216-t003:** Biomarkers and histological characteristics during follow-up of overall KTRs and randomization groups.

Biomarkers and Histological Characteristics	Overall	Randomized Group		*p*
		CsA	EVL	
**Biomarkers**				
**6 Months**				
*Plasma*				
PRO-C6 (ng/mL) ^a^	9.5 (3.4)	9.5 (3.1)	9.4 (3.9)	0.93
Creatinine, µmol/L (SD)	130 (33)	130 (31)	130 (35)	0.96
*Urine*				
PRO-C6 (ng/mg creat) ^b^	6.7 (4.8–12.4)	6.6 (4.9–12.9)	6.8 (3.8–12.8)	0.70
**24 Months**				
*Plasma*				
PRO-C6 (ng/mL) ^c^	9.4 (4.3)	9.6 (4.5)	9.1(4.3)	0.72
Creatinine, µmol/L (SD)	143 (49)	149 (46)	136 (53)	0.22
*Urine*				
PRO-C6 (ng/mg creat) ^b^	5.9 (3.4–21.5)	4.5 (3.2–10.2)	7.5 (4.6–40.7)	0.02
**Delta_24-6_**				
*Plasma*				
PRO-C6 (ng/mL) ^c^	0.3 (3.9)	0.6 (3.1)	0.01 (4.6)	0.67
*Urine*				
PRO-C6 (ng/mg creat) ^b^	−0.5 (−2.6–4.8)	−1.4 (−3.6–0.27)	0.9 (−2.2–23.9)	0.01
**Histological analyses**				
**6 Months**				
IF/TA-score	1 (0–1)	1 (0–1)	1 (1–2)	0.56
PSR, %	13.3 (6.0)	13.0 (6.1)	13.6 (6.0)	0.65
ti-score, %	10.0 (5.0–15.8)	10.0 (5.0–10.0)	10.0 (5.0–20.0)	0.38
**24 Months**				
IF/TA-score	1 (1–2)	1 (0–1)	1 (1–2)	0.36
PSR, %	17.3 (10.6)	19.7 (11.7)	14.5 (8.5)	0.02
ti-score, %	20.0 (10.0–41.3)	20.0 (10.0–50.0)	15.0 (10.0–30.0)	0.16
**Delta_24-6_**				
IF/TA-score	0.5 (0–1)	1 (0–1)	0 (0–1)	0.23
PSR, %	4.0 (11.4)	6.7 (13.1)	0.9 (7.9)	0.01
ti-score, %	10 (0–30)	10 (0–45)	5 (0–20)	0.09

Data available in ^a^ 73, ^b^ 62, ^c^ 36 patients. CsA: cyclosporine group; EVL: everolimus group; PRO-C6: released pro-peptide of collagen type VI (endotrophin); IF/TA: interstitial fibrosis/tubular atrophy; PSR: Picro Sirius Red; ti-score: total inflammation score.

**Table 4 jcm-09-03216-t004:** Association of histological analyses with plasma and urine PRO-C6.

Histological Analyses	6-Months PRO-C6		24-Months PRO-C6		Delta_24-6_ PRO-C6	
	Plasma, ng/mL	Urine, ng/mg	Plasma, ng/mL	Urine, ng/mg	Plasma, ng/mL	Urine, ng/mg
	Std. β	Std. β	Std. β	Std. β	Std. β	Std. β
**6 Months**						
IF/TA	0.34 **	0.20				
PSR	0.11	−0.18				
ti-score	0.04	0.08				
**24 Months**						
IF/TA	0.24 *	0.06	0.08	0.13	−0.04	0.02
PSR	0.01	−0.30 *	0.06	−0.24	−0.01	0.04
ti-score	0.23 *	0.23	0.16	0.09	0.05	−0.03
**Delta_24-6_**						
IF/TA	−0.03	−0.08	−0.20	−0.07	−0.16	0.01
PSR	−0.06	−0.17	−0.009	−0.19	0.04	−0.04
ti-score	0.22 *	0.20	0.11	−0.02	0.01	−0.02

* *p* value < 0.05; ** *p* value < 0.01. Linear regression analyses were performed. Std. β coefficients represent the difference (in standard deviations) in each biomarker per 1 standard deviation increment in each individual biopsy score. PRO-C6: pro-peptide of collagen VI (endotrophin); Std. β: standardized beta coefficient; IF/TA: interstitial fibrosis/tubular atrophy; PSR: Picro Sirius Red; ti-score: total inflammation score.

**Table 5 jcm-09-03216-t005:** Association of histological analyses with plasma and urine PRO-C6 among CsA and EVL groups.

Histological Analyses	24 Months				Delta_24-6_			
	Plasma PRO-C6, ng/mL		Urine PRO-C6, ng/mg		Plasma PRO-C6, ng/mL		Urine PRO-C6, ng/mg	
	CsA	EVL	CsA	EVL	CsA	EVL	CsA	EVL
	Std. β	Std. β	Std. β	Std. β	Std. β	Std. β	Std. β	Std. β
**24 Months**								
IF/TA	−0.11	0.32	−0.25	0.30	−0.31	0.16	0.13	−0.11
PSR	−0.07	0.18	−0.09	−0.31	−0.47	0.25	0.12	−0.11
ti-score	−0.07	0.40	0.18	0.14	−0.37	0.32	0.08	−0.07
**Delta_24-6_**								
IF/TA	−0.30	−0.07	−0.43 *	0.07	−0.40	−0.01	0.12	−0.003
PSR	0.001	−0.06	−0.10	−0.27	−0.28	0.25	0.03	−0.18
ti-score	−0.08	0.33	0.07	0.05	−0.35	0.25	0.09	0.04

* *p* value < 0.05. Linear regression analyses were performed. Std. β coefficients represent the difference (in standard deviations) in each biomarker per 1 standard deviation increment in each individual biopsy score. PRO-C6: pro-peptide of collagen type VI (endotrophin); CsA: cyclosporine group; EVL: everolimus group; Std. β: standardized beta coefficient; IF/TA: interstitial fibrosis/tubular atrophy; PSR: Picro Sirius Red; ti-score: total inflammation score.

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
