# Peer review of "Biopsy-Controlled Non-Invasive Quantification of Collagen Type VI in Kidney Transplant Recipients: A Post-Hoc Analysis of the MECANO Trial"

_jcm, 2020, doi:10.3390/jcm9103216_

Round 1

Reviewer 1 Report

Gn recurrence in transplanted patients remains a concerning topic: indeed the low rate of recurrence is, at least in part, affected by the low use of IF studies in these patients. The results probably underestimate the real phenomenon. In your study, 17% of patients have a previous diagnosis of primary GN. The same percentage for primary kidney disease is classified as "other". Are these patients biopsed or not biopsied? in the latter condition, is it for late referall or is there a mix of late referral plus genetic disorders? If these percentage are all not biopsed patients, near 1/3 of collected patients could potentially have a GN not evaluated at the time of transplant biopsy: does the statistical impact remain the same? how many patients with no primary diagnosis supported by a renal biopsy did you have?
Gn recurrence is vary as Authors reported, and the percentage reported by Lim et al is limited to primary GN, they did not focus on secondary GN (e.g. lupus nephritis, infectious-related, cryo-GN, MGRS, pauci-immune GN..).

Serum creatinine evolution is a key point in order to keep monitoring renal function in the first year of transplant and to catch functional or structural abnormalities. The lack of serum creatinine evaluation at time zero is a significant missing point that negatively influenced 6 month biopsy (DGF versus subclinical rejection, versus functional CNI, other?) to explain early fibrotic evolution.

I would suggest to add these missing points and eventually comparing PRO-C6 pre and post transplantation as well in order to improve the quality of this interesting study.

Author Response

We thank the reviewer for the comments on our work. Please see the attachment for a detailed response.

Reviewer 2 Report

The manuscript can be accepted in the current version. 

Author Response

We thank the reviewer for the kind appraisal of our work.

Round 2

Reviewer 1 Report

I do not have more suggestions or comments to add. I appreciate the effort to highlight paper strengths and weakness as well

This manuscript is a resubmission of an earlier submission. The following is a list of the peer review reports and author responses from that submission.

Round 1

Reviewer 1 Report

This is an interesting post hoc analysis of a non invasive test for collagen and its correlation with protocol biopsy findings.  The paper is well written.  The findings are statistically significant, but at a low level, and it is unclear that this non invasive test has important predictive value.  Still, the field of immunologic and histopathologic markers is sufficiently poorly developed that this paper should be published.  No revisions from my point of view.  The editors should check with a statistical editor to make sure that no additional analyses need to be performed.

Reviewer 2 Report

Yepes-Calderon and colleagues describe the correlation between 6-months post-transplant plasma concentration of PRO-C6 and graft biopsy-proven fibrotic and inflammatory changes (IFTA score and Ti score) proposing it as a possible future non-invasive biomarker for fibrosis development.

It is reported that increased plasma levels of PRO-C6 have been linked with the progression of chronic kidney disease, also in the post-transplantation setting. Few studies have investigated whether associations between PRO-C6 and decreased graft function indeed correspond to increased fibrotic or inflammatory changes in the kidneys.

The paper is well written and structured with a homogeneous well-characterized cohort of KTR. It represents a useful article for researchers who want to deepen the role of new biomarkers for kidney transplants.

As reported at the end of the discussion, authors describe limitations of this hypothesis:  plasma or urine PRO-C6 level, only measures a collagen split product of the alpha3 chain of collagen VI, whereas PSR staining is the resultant of all collagen deposits. also, PRO-C6 assay measures a split product of collagen VI which is cleaved off after cellular synthesis and thus reflects the synthesis of collagen VI instead its deposition in a tissue is the result of collagen synthesis and collagen degradation.

Some minor revisions are needed:

  • please review the reference numbers, because there is a mismatch between the references ( Reference number 26 (Mengel, M.et al; Scoring Total Inflammation Is Superior to the Current Banff Inflammation Score in Predicting 404 Outcome and the Degree of Molecular Disturbance in Renal Allografts. Am. J. Transplant. 2009) is not reported in the article).
  • please on page 3 line 120 -122 it is not clear PCR staining (the specific role for the collagen detection)

Reviewer 3 Report

This is a well presented study with significant results. It is relevant and hopefully indeed a step forward.

Reviewer 4 Report

The Authors investigated in a post-hoc analysis whether monitoring PRO-C6 in plasma and urine in MACANO trials’ KTR could be a reliable option in catching renal chronic fibrotic decline. They design a post-hoc analysis based on histology and urine and blood samples at 6 and 24 months after transplantation.

Some questions rise:

  1. Were patients with previous diagnosis of glomerulonephritis excluded in case of recurrent GN?
  2. The Authors did not mention immunofluorescence studies, how are you sure you did or did not have recurrent GN? Perhaps, could interstitial inflammation be secondary to a recurrent GN?
  3. Did the Authors find any differences in stratifying patients for their primary kidney disease?
  4. The Authors explain very well Collagen VI role in “healthy” and “pathologic” circumstances. How we consider transplanted patients that reach normal kidney function? What about have a control group with ESRD patients for testing serum PRO-C6 and comparing results with KTR? Or even better, to test PRO-C6 in patients before and after transplantation?
  5. How were patients’ medium serum creatinine at the beginning, at 6 and 24 months? Indeed, eGFR seems a little bit low in a restoring renal condition as KTR are supposed to be.

I had some difficulties to appreciate the benefit in dosing PRO-C6 in KTR because results were inconstant and not “linear” and uniform as a marker should be (i.e: serum PRO-C6 increasing over time mirroring the kidney aging and so fibrosis; serum PRO-C6 should be associated with interstitial inflammation as the Author explain in the discussion and I expected more correlation with the lesser inter-observer variation PSR technique).

Reviewer 5 Report

In this valuable research the authors aimed to investigate the connection between PRO-C6 levels and fibrosis status in kidney biopsies of renal transplant. recipients.

The design of this study is well written and the research is well performed. 
The findings of this research will improve our knowledge of fibrosis markers and are in line with the purpose of the journal. 

Only minor spell and form modification are needed: 

At line 86 changes "regimes" into "regimens".

At line 246 you have two independent clauses improperly joined with a comma. Consider correcting the comma splice "Under healthy conditions it has an important physiological role in maintaining structure and function, and controlling the organization of the ECM and cell orientation, however, its marked increased synthesis and deposition has been reported under a wide spectrum of kidney pathologies".

Line 272 the noun collagen might combine better with an adjective other than actual: change "actual" with "existing".

Line 298 change "This study for the first-time provides biopsy299 controlled data of PRO-C6 as potential non-invasive biomarker of graft fibrosis in KTR". into "For the first time, this study provides..."